# Exceptional uranium(VI)-nitride triple bond covalency from 15N nuclear magnetic resonance spectroscopy and quantum chemical analysis

Jingzhen Du [1], John A. Seed [1], Victoria E. J. Berryman[1], Nikolas Kaltsoyannis[1], Ralph W. Adams [1✉], Daniel Lee [2✉] & Stephen T. Liddle [1✉]

Determining the nature and extent of covalency of early actinide chemical bonding is a fundamentally important challenge. Recently, X-ray absorption, electron paramagnetic, and nuclear magnetic resonance spectroscopic studies have probed actinide-ligand covalency, largely confirming the paradigm of early actinide bonding varying from ionic to polarised-covalent, with this range sitting on the continuum between ionic lanthanide and more covalent d transition metal analogues. Here, we report measurement of the covalency of a terminal uranium(VI)-nitride by 15N nuclear magnetic resonance spectroscopy, and find an exceptional nitride chemical shift and chemical shift anisotropy. This redefines the 15N nuclear magnetic resonance spectroscopy parameter space, and experimentally confirms a prior computational prediction that the uranium(VI)-nitride triple bond is not only highly covalent, but, more so than d transition metal analogues. These results enable construction of general, predictive metal-ligand 15N chemical shift-bond order correlations, and reframe our understanding of actinide chemical bonding to guide future studies.

[1] Department of Chemistry, The University of Manchester, Oxford Road, Manchester M13 9PL, UK. [2] Department of Chemical Engineering and Analytical Science, The University of Manchester, Manchester M13 9PL, UK. ✉email: ralph.adams@manchester.ac.uk; daniel.lee@manchester.ac.uk; steve.liddle@manchester.ac.uk

Determining the nature and extent of covalency, that is the extent of electron sharing between two elements, in actinide-ligand (An–L) bonding is a central and enduring fundamental goal of actinide science[1–3]. For this reason, the study of An–L multiple bonding is of burgeoning interest since such bonds tend to inherently exhibit levels of covalency that are practical to investigate[4–9]. The generally accepted chemical bonding picture is that for early actinides the bonding varies from ionic to polarised-covalent as a function of An-oxidation state and ligands, with this range sitting intermediate to the ionic lanthanides and usually much more covalent $d$ transition metal complexes[5,7,8,10,11]. Probing actinide covalency is challenging, but in recent years progress has been made using experimental approaches, underpinned by quantum chemical calculations, including K-edge X-ray absorption near edge spectroscopy[12–18], pulsed electron paramagnetic resonance spectroscopy[19], and nuclear magnetic resonance (NMR) spectroscopy[20–36]. These investigations have begun to place the bonding descriptions of An–L bonding on a rigorous, quantitative footing, and have been consistent with the status quo bonding description of early actinides.

Regarding the use of NMR spectroscopy to probe An–L covalency, solution, and solid-state studies encompassing $^1$H[20], $^{13}$C[21–24], $^{15}$N[25,26], $^{17}$O[27–30], $^{19}$F[31–34], $^{35/37}$Cl[35], $^{77}$Se[36], and $^{125}$Te[36] nuclei have revealed that their chemical shift properties are highly sensitive to interactions with actinide ions, thus constituting a direct, powerful experimental probe of chemical bonding and covalency when the individual bonding contributions to shielding tensors are analysed in detail[20–22,24–26,36,37]. A potential benchmark at the interface of NMR spectroscopic and An–L multiple bond investigations is the terminal uranium(VI)-nitride triple bond, due to its formally closed-shell diamagnetic formulation rendering it amenable to study by NMR spectroscopy. However, despite the sustained nature of An–L multiple bond chemistry, terminal actinide-nitrides remain rare, with only two classes of isolable terminal uranium-nitride reported. In 2012, the Liddle group reported $[U^V(N)(Tren^{TIPS})][Na(12C4)_2]$ ($Tren^{TIPS} = N(CH_2CH_2NSiPr^i_3)_3$, 12C4 = 12-crown-4 ether)[38], and in 2013 disclosed its oxidation to give $[U^{VI}(N)(Tren^{TIPS})]$ (**1**)[39]. In 2020 the Mazzanti group[40] reported $[U^{VI}(N)\{OSi(O-Bu^t)_3\}_4][NBu^n_4]$.

Prior quantum chemical modelling of **1** suggested a highly covalent U≡N triple bond rivalling, if not exceeding, the covalency in group 6 nitride triple bonds[39,41], which would be a significant result if experimentally confirmed given the dominance of 5$f$- over 6$d$-orbital participation in this bond, and exceptional chemical shielding tensors have thus been predicted for uranium-nitride linkages[25]. While some comparative uranium(V/VI)-nitride reactivity studies have begun to build a picture consistent with the view that the uranium(VI)-nitride linkage is highly covalent[42–44], definitive experimental spectroscopic confirmation of the computational description of high covalency that exceeds $d$

transition metal analogues has remained lacking. However, it has previously been shown to be possible to prepare the $^{15}$N$_{nitride}$ enriched $[U^{VI}(N^*)(Tren^{TIPS})]$ (**1***, N* = 50:50 $^{14}$N:$^{15}$N)[39], presenting an opportunity for benchmarking NMR studies to experimentally test the above predictions. Indeed, to the best of our knowledge only one other actinide-nitride complex has been studied by $^{15}$N NMR spectroscopy, namely the bridging dithorium-nitride complex $[(N'')_3Th(\mu-N)Th(N'')_3][K(18C6)(THF)_2]$ ($N'' = N(SiMe_3)_2$) described by the Hayton, Autschbach, and Cho groups[26].

Here, we report a solution- and solid-state $^{15}$N NMR spectroscopic study of the terminal actinide-nitride complex **1***. We find exceptional $^{15}$N chemical shift and chemical shift anisotropy that to the best of our knowledge redefine the range for N-containing compounds. We find that these $^{15}$N properties are dominated by the paramagnetic shielding term, with spin–orbit contributions being minor. The $^{15}$N NMR data experimentally confirm that the U≡N triple bond is highly covalent, and indeed more so than group 4–6 transition metal nitrides. This reframes our understanding of the nature and range of the covalency of An–L chemical bonding, and permits construction of metal-nitride $^{15}$N chemical shift correlated to several quantum chemical measures of bond order as general predictive models.

## Results

**Synthesis of 1*.** We prepared a sample of 50% $^{15}$N$_{nitride}$ enriched **1*** according to Fig. 1. This is a modified method to the previously reported preparation of **1*** (refs. [39,41,42]), where use of $[K(B15C5)_2]^+$ (B15C5 = benzo-15-crown-5 ether) instead of $[Na(12C4)_2]^+$ as the cation component of the $[U(^{14/15}N)(Tren^{TIPS})][M(L)_2]$ (M = Na, L = 12C4; M = K, L = B15C5) precursor to **1*** is found to be more reliable and thus practicable. The purity and stability of **1*** was checked and confirmed by $^1$H and $^{29}$Si NMR spectroscopies in D$_8$-THF, 50:50 D$_8$-THF:C$_6$D$_6$, and C$_6$D$_6$ (Supplementary Figs. 1–4).

**Solution and solid-state $^{15}$N NMR spectroscopic investigation of 1*.** Previously, we were unable to detect the $^{15}$N$_{nitride}$ resonance for **1***[39], attributed to the low molar receptivity (~0.1% of $^1$H) and high longitudinal relaxation time constant of the $^{15}$N nucleus. However, using a low flip angle acquisition (10° r.f. pulse, 1 s acquisition, 1.7 s recycle delay), we were able to clearly observe the $^{15}$N$_{nitride}$ resonance of **1***. The $^{15}$N{$^1$H} NMR spectrum of **1*** in D$_8$-THF (Fig. 2a) exhibits a single isotropic chemical shift ($\delta_{iso}$) resonance at 968.9 ppm relative to the IUPAC standard of MeNO$_2$ at 0 ppm[45]. The analogous spectrum of **1*** in D$_6$-benzene exhibits a resonance at $\delta_{iso}$ 972.6 ppm, demonstrating little polar/non-polar solvent medium dependence of the $^{15}$N$_{nitride}$ $\delta_{iso}$ of **1***.

Few molecular metal nitrides have been characterised by $^{15}$N NMR spectroscopy in solution, and while some have been referenced to MeNO$_2$ = 0 ppm, others are referenced to NH$_{3(l)}$ = 0 that would

**Fig. 1 Multi-step synthesis of 1*.** The initial chloride complex is converted to the corresponding azide by salt elimination. Reduction releases N$_2$ to produce a bridging nitride dimer, which can be cleaved into a separated ion pair by treatment with the appropriate crown ether. Oxidation of the separated ion pair gives neutral, 50% $^{15}$N-labelled **1*** for the NMR investigations described in this study. B15C5 = benzo-15-crown-5 ether.

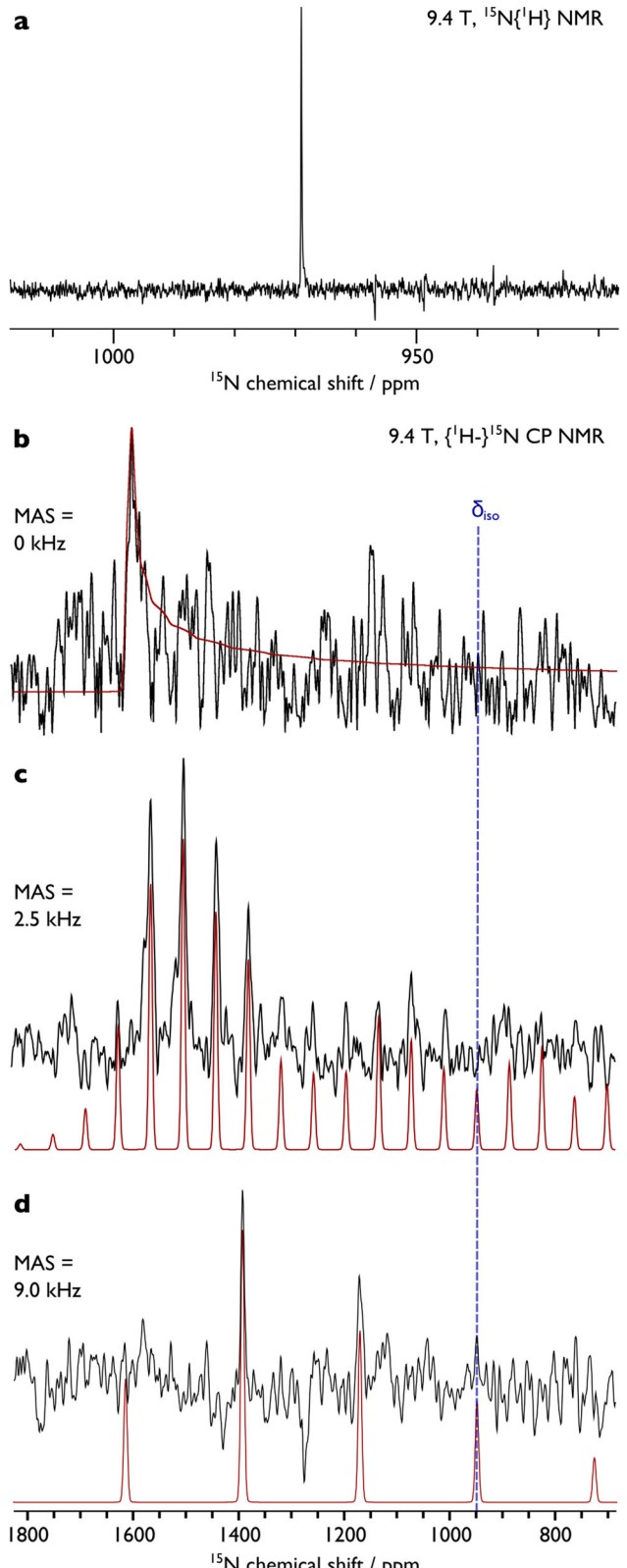

**Fig. 2 $^{15}$N NMR spectra of 1\* recorded at 298 K. a** Solution spectrum in D$_8$-THF. **b** Solid-state spectrum with MAS = 0 kHz. **c** Solid-state spectrum with MAS = 2.5 kHz. **d** Solid-state spectrum with MAS = 9.0 kHz. In each case the black line is the experimental spectrum, and the red line is the simulated spectrum.

make MeNO$_2$ = 380.2 ppm under the measurement conditions used; in the latter reference frame the $^{15}$N$_{nitride}$ $\delta_{iso}$ of 1\* is then 1349.1/1352.8 ppm, respectively. Accordingly, we list pertinent nitrides with both associated $\delta_{iso}$ values (MeNO$_2$/NH$_{3(l)}$ = 0 ppm): 1\* ($\delta_{iso}$ = av. 971/1351 ppm); [V(N)(L$^{MeDipp}$)(ODipp)] ($\delta_{iso}$ = 679/1059 ppm, L$^{MeDipp}$ = HC(CMeNDipp)$_2$, Dipp = 2,6-Pr$^i_2$C$_6$H$_3$)[46]; [V(N)(L$^{Me-Dipp}$)(NTol$_2$)] ($\delta_{iso}$ = 669/1049 ppm, Tol = 4-Me-C$_6$H$_4$)[47]; [V(N)(L$^{MeDipp}$){N(Tol)(Mes)}] ($\delta_{iso}$ = 655/1035 ppm, Mes = 2,4,6-Me$_3$C$_6$H$_2$)[47]; [Ti(N)(NP)$_2$][K(2,2,2-crypt)] ($\delta_{iso}$ = 578/958 ppm, NP = MesNC$_6$H$_3$-3-Me-2-PPr$^i_2$)[48,49]; [Ti(μ-N)(NP)$_2$K(18C6)] ($\delta_{iso}$ = 542/922 ppm)[48,49]; [Ti(μ-N)(NP)$_2$K(OEt$_2$)]$_2$ ($\delta_{iso}$ = 512/892 ppm)[48,49]; [Mo(N){N(Bu$^t$)(C$_6$H$_3$-3,5-Me$_2$)$_3$}] ($\delta_{iso}$ = 460/840 ppm)[50]; [(N″)$_3$Th(μ-N)Th(N″)$_3$][K(18C6)(THF)$_2$] ($\delta_{iso}$ = 299/679 ppm)[25]. Thus, the $^{15}$N$_{nitride}$ resonance for 1\* is the most downfield (highest frequency) deshielded $\delta_{iso}$ value to date, extending the known $^{15}$N NMR $\delta_{iso}$ range by ~300 ppm.

Given the downfield $^{15}$N$_{nitride}$ $\delta_{iso}$ value of 1\*, we recorded its solid-state NMR spectrum, Fig. 2b–d. The spectrum of the static sample is broad, but the general shape of the spectrum can be discerned. Under magic angle spinning (MAS) conditions, an improved signal-to-noise ratio is obtained and rotational side-bands are well-defined at 2.5 kHz spinning frequency that highlight a chemical shift anisotropy of ~2000 ppm. The signal-to-noise ratio is further improved at a MAS frequency of 9.0 kHz, but now the frequency-dependent rotational side-bands extend beyond the available pulse bandwidth. Nevertheless, taken together these data enable reliable spectral simulation, yielding a $\delta_{iso}$ value of 950 ppm, which is in good agreement with the solution $\delta_{iso}$ value ($\Delta_{sol-ss}$ = 21 ppm), and chemical shift tensor $\delta_{xx}$, $\delta_{yy}$, $\delta_{zz}$, $\Omega$ (tensor span, $\delta_{xx} - \delta_{zz}$), and $\kappa$ (skew, $[3(\delta_{yy} - \delta_{iso})]/\delta_{xx} - \delta_{zz}$) values of 1617, 1603, −370, 1987 ppm, and 0.99, respectively.

The $\Omega$ value for 1\* is notably large, and can be compared to those of [Ti(N)(NP)$_2$][K(2,2,2-crypt)] ($\Omega$ = 1470 ppm)[48,49], [Mo(N){N(Bu$^t$)(C$_6$H$_3$-3,5-Me$_2$)$_3$}] ($\Omega$ = 1187 ppm)[50], [Ti(μ-N)(L$^{ButDipp}$)(NTol$_2$)K]$_2$ ($\Omega$ = 1155 ppm)[51], and [(N″)$_3$Th(μ-N)Th(N″)$_3$][K(18C6)(THF)$_2$] ($\Omega$ = 847 ppm)[26]. Thus, though there are comparatively few data for comparison, 1\* exhibits the largest $\Omega$ of any nitrogen-containing molecule, enlarging the known tensor span range by ~500 ppm. Like most other molecular metal nitrides, the $\kappa$ value for 1\* is close to one, and the line shape of the spectra are characteristic of an axially symmetric shift tensor. These findings are consistent with the structure of the uranium-nitride linkage in 1\*, which is highly axial[41], residing along a three-fold symmetry axis defined by Tren$^{TIPS}$.

**Computational benchmarking of the $^{15}$N NMR spectroscopic properties of 1\*.** The $^{15}$N$_{nitride}$ NMR spectroscopic data for 1\* prompted us to computationally model its NMR properties in detail. We used coordinates from geometry optimisation (see Supplementary Table 1) at the BP86 level[39], noting the computed U ≡ N bond length of 1.7795 Å compares well to the crystallographically determined distance of 1.799(7) Å[39]. To give context to the U ≡ N bond length distance in 1/1\*, the corresponding distances in [U$^V$(N)(Tren$^{TIPS}$)][Na(12C4)$_2$] and [U$^{VI}$(N){OSi(OBu$^t$)$_3$}$_4$][NBu$^n_4$] are 1.825(15) and 1.769(2) Å, respectively[38,40]. We then calculated $^{15}$N$_{nitride}$ $\delta_{iso}$ values from single point energy calculations with a range of functionals, corrected for the solvent (THF, since that is the most accurately determined $\delta_{iso}$ for 1\*) and reference (MeNO$_2$ = 0 at the same functional level). We found (Supplementary Table 2), irrespective of using scalar relativistic (SR) or two-component spin–orbit

 **3**

relativistic (SOR) effects, that BP86 and SAOP functionals underestimate the $\delta_{iso}$ value ($\Delta_{calc-exp} = -166$ to $-359$ ppm) whereas PBE0 and PBE0-HF40% over-estimate the $\delta_{iso}$ value ($\Delta_{calc-exp} = +150$ to $+401$ ppm). However, the B3LYP functional gave better agreement, with $\delta_{iso}$ at 1040 ppm (scalar) or 1044 ppm (spin–orbit) ($\Delta_{calc-exp} = \sim+70$ ppm). Inclusion of dispersion did not improve the level of agreement, because optimised geometries returned unrealistically short $U \equiv N$ bond lengths suggesting over-compensated dispersion, so we adopted an established empirical correction approach (Supplementary Table 3 and Supplementary Fig. 5, $R^2 = 0.9951$)[52,53], giving a corrected computed $\delta_{iso}$ value of 966 ppm ($\Delta_{calc-exp} = -5$ ppm). To provide support for the computed $^{15}N_{nitride}$ $\delta_{iso}$ of $\mathbf{1}^*$, we calculated the $^{29}Si$ NMR $\delta_{iso}$ for the $SiPr^i_3$ groups in $\mathbf{1}^*$ at the B3LYP level, and find a value of $-0.61$ ppm, in good agreement with the experimental $\delta_{iso}$ of 3.8 ppm in THF[54].

Focussing on the B3LYP two-component SOR model, we calculated $N_{amide}$ and $N_{amine}$ $\delta_{iso}$ values of $-123$ and $-311$ ppm, respectively. for the $Tren^{TIPS}$ ligand in $\mathbf{1}^*$. The computed $N_{amide}$ $\delta_{iso}$ of $\mathbf{1}^*$ agrees well with the reported $^{15}N$ NMR $\delta_{iso}$ of $-198$ ppm for the $Th-NH_2$ unit in $[Th(NH_2)(N'')_3]$[25], and we note that the computed $N_{nitride}$, $N_{amide}$, and $N_{amine}$ $\delta_{iso}$ values for $\mathbf{1}^*$ are positively correlated (Supplementary Figs. 6 and 7, $R^2 = 0.9983$ or $0.9988$) when plotted against their U–N Mayer bond orders (2.94, 0.91, 0.44) or U–N delocalisation indices (DI, 2.649, 0.790, 0.377), respectively. As expected, there is little variation in the computed diamagnetic isotropic shielding tensors ($\sigma^d$) for the $N_{nitride}$ (320 ppm), $N_{amide}$ (311 ppm), and $N_{amine}$ (309 ppm) centres, since $\sigma^d$ derives principally from tightly-bound core electron densities[55]. In contrast, the paramagnetic isotropic shielding tensors ($\sigma^p$) show substantial variation, being $-1433$ ($N_{nitride}$), $-363$ ($N_{amide}$), and $-186$ ($N_{amine}$) ppm, respectively, reflecting that $\sigma^p$ derives from polarisable valence electron densities that directly report different chemical bonding environments arising from magnetically induced-mixing of the ground state with excited states[55]. Interestingly, the SO shielding tensors ($\sigma^{so}$) are relatively small, being $-29$ ($N_{nitride}$), $-40$ ($N_{amide}$), and $-20$ ($N_{amine}$) ppm.

With the $^{15}N_{nitride}$ solution $\delta_{iso}$ value of $\mathbf{1}^*$ satisfactorily reproduced computationally, we analogously modelled (Supplementary Table 4 and Supplementary Fig. 8, $R^2 = 0.9977$) the solid-state data, computing a solid-state $^{15}N_{nitride}$ $\delta_{iso}$ of 935 ppm ($\Delta_{calc-exp} = -15$ ppm). The computed $\sigma^p$, $\sigma^d$, and $\sigma^{so}$ values ($-1492$, 318, and $-27.0$ ppm, respectively) are similar to the computed solution parameters; also, the computed $\delta_{xx}$, $\delta_{yy}$, $\delta_{zz}$, $\Omega$, and $\kappa$ values of 1590, 1584, $-378$, 1969 ppm, and 0.99 are in excellent agreement with the simulation of the experimental solid-state spectra. These data suggest that the modest difference in solution and solid-state $\delta_{iso}$ values are due to solvent effects, and that fast dynamic averaging effects do not operate for $\mathbf{1}^*$ in the solid-state at ambient temperature.

**Computational electronic structure analysis of $\mathbf{1}^*$.** Having established that the B3LYP functional appropriately models the experimental $^{15}N$ NMR data for $\mathbf{1}^*$, we re-analysed the electronic structure of $\mathbf{1}^*$ in order to rationalise its downfield $^{15}N_{nitride}$ $\delta_{iso}$. Previously reported calculations on $\mathbf{1}^*$ using the BP86 functional[39] revealed the the highest occupied molecular orbital (HOMO) to be the $U \equiv N$ $\sigma$-bond, the HOMO-1 and -2 are $U \equiv N$ $\pi$-bonds, then resides a manifold of U–$N_{amide}$ bonding combinations. B3LYP calculations on $\mathbf{1}^*$ reveal essentially the same gross $U \equiv N$ $\sigma > \pi$ energy bonding description, but now the $U \equiv N$ combinations are more greatly delocalised with the U–$N_{amide}$ manifold, resulting in three groups of three MOs each, with each triad exhibiting varying orbital coefficient weightings of

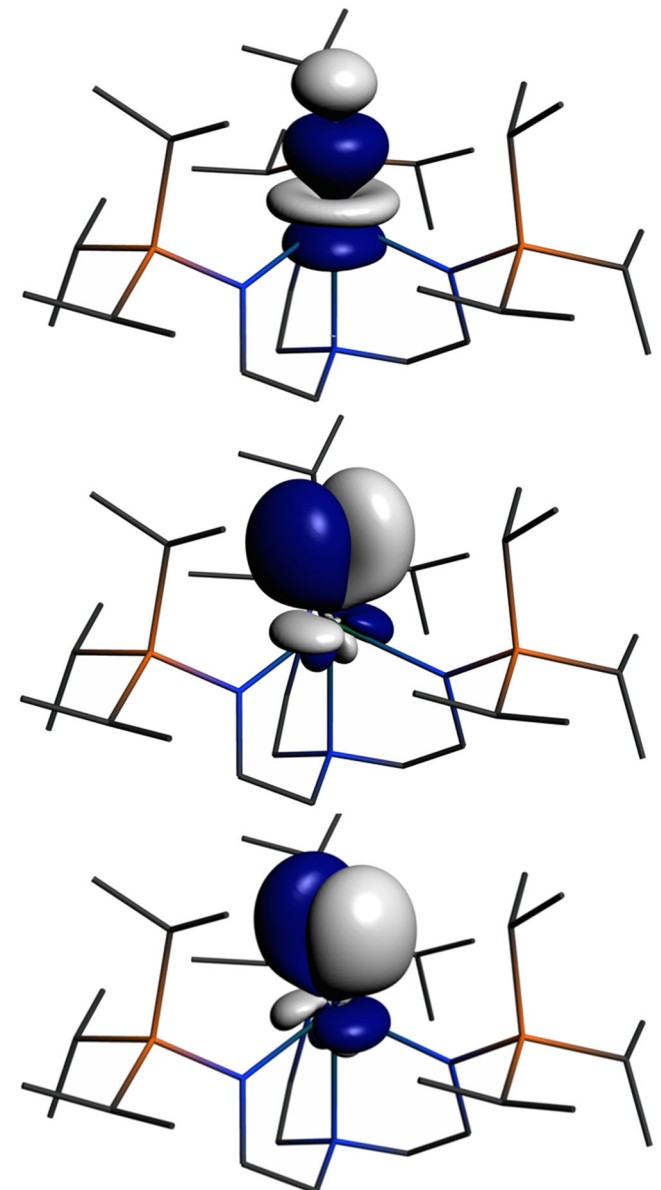

**Fig. 3 Natural Localised Molecular Orbitals (NLMOs) of $\mathbf{1}^*$.** These three NLMOs represent the $U \equiv N$ $\sigma$- and $\pi$-bonds. Hydrogen atoms are omitted for clarity.

the $U \equiv N$ $\sigma \rightarrow \pi$-combinations mixed in with U–$N_{amide}$ combinations. However, the principal $U \equiv N$ $\sigma$- and $\pi$-bonds can be identified as HOMO$-3$ to $-5$.

In order to provide a chemically more intuitive model, we examined the NLMOs of $\mathbf{1}^*$ at the B3LYP level (Fig. 3) using NBO6. The $U \equiv N$ $\sigma$-bond is 39.9% U ($7s:7p:6d:5f = 3:1:10:86\%$) and 58.9% N ($2s:2p = 8:92\%$) character, whereas the $U \equiv N$ $\pi$-bonds are 31% U ($7s:7p:6d:5f = 0:0:24:76\%$) and 68% N ($2s:2p = 0:99\%$). The B3LYP NLMO (NBO6) data are in good agreement with previously reported BP86 NBO (NBO5) analysis of $\mathbf{1}^*$[39], and emphasise the highly covalent nature, and dominance of $5f$ U and $2p$ N orbitals, in the bonding of the $U \equiv N$ linkage in $\mathbf{1}^*$.

**Computational chemical shift analysis of $\mathbf{1}^*$.** Ramsey's formula, Eq. (1), examines NMR interactions from a quantum–mechanical perspective, decomposing magnetic shielding contributions into $\sigma^d$ and $\sigma^p$ components that are dependent on electron orbital

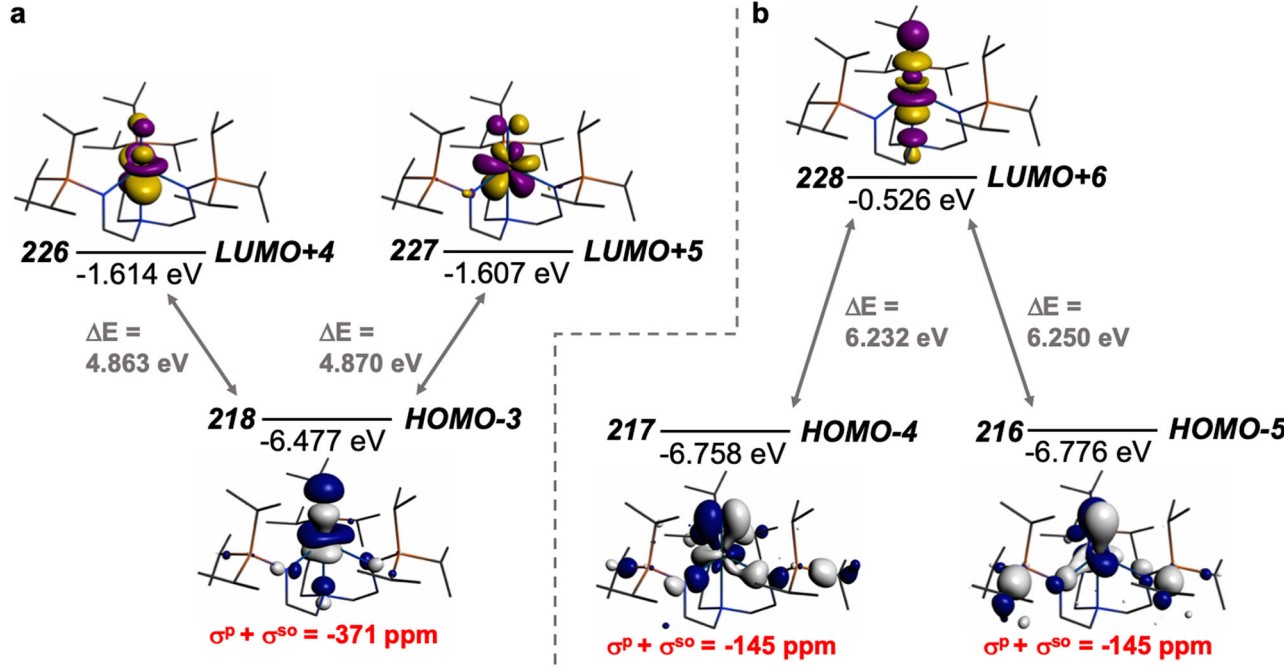

**Fig. 4 Dominant occupied and virtual molecular orbitals that contribute to the $\delta_{iso}$ value of 1\* by magnetic field-induced mixing. a** Mixing of the occupied U≡N σ-bond with unoccupied U≡N π\*-bonds. **b** Mixing of the occupied U≡N π-bonds with unoccupied U≡N σ\*-bond. Hydrogen atoms are omitted for clarity.

angular momenta[56–58]. While this does not directly translate to the MO approach of hybrid DFT, it provides a framework with which to rationalise NMR magnetic shieldings using DFT B3LYP calculations when SO contributions are included, Eq. (2)[44].

$$\sigma_{iso} = \sigma^d + \sigma^p. \tag{1}$$

$$\sigma_{iso} = \sigma^d + \sigma^p + \sigma^{so}. \tag{2}$$

Since for 1\* $\sigma^d$ varies little for any of the N-environments and $\sigma^{so}$ is fairly small, we focus on $\sigma^p$. This term can be presented in reduced form as:

$$\sigma^p \propto \frac{1}{r^3} \times \frac{\sum_{M} Q_{NM}}{\Delta E}. \tag{3}$$

Here, $r$ is the radial expansion of the shielding electrons from the nucleus being examined, $N$ denotes the NMR nucleus, $\Sigma_M Q_{NM}$ is the sum of the charge density and bond order matrix elements over the relevant atoms (M), and $\Delta E$ is the energy separation between the ground and excited states in question[36,37,45,56–58]. It follows that $\sigma^p$ is proportional to the magnetic field-induced mixing of magnetically coupled orbitals and inversely proportional to $r^3$. For the latter this arises because as a nucleus (M) withdraws charge from the NMR nucleus (N) the valence orbitals of N will contract so the $1/r^3$ term will become larger (more deshielded NMR nucleus) resulting in a larger $\sigma^p$ term. Put another way, the larger the bond order of, so more covalent, the bond involving the NMR nucleus the larger $\sigma^p$ will become[36]. The $\sigma^p$ term is also inversely proportional to the energy gap between the occupied and virtual orbitals that become magnetically coupled, so smaller $\Delta E$ gaps produce larger $\sigma^p$ values. We note in passing that the HOMO-LUMO gap of 1\* is unremarkable (see Supplementary Table 5), and so would be expected to be subordinate to the $1/r^3$ term in terms of its contribution to $\sigma^p$. The mixing by magnetic coupling of occupied and virtual orbitals must be symmetry allowed, since the angular momentum operators belong to the same irreducible representations as the rotational operators. Canonical MOs are often

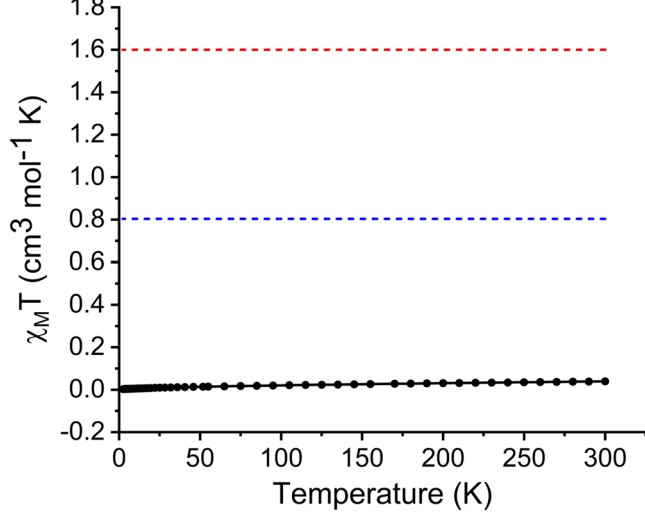

**Fig. 5 SQUID $\chi_M T$ (cm³ mol⁻¹ K) vs $T$ (K) magnetic data for 1\* in an external 1 T field.** Experimental data points are represented by black circles and the line is a guide to the eye only. For comparison, the hypothetical magnetic moments of $5f^1$ and $5f^2$ ions are represented by the blue and red dotted lines, respectively.

delocalised, and so contributions to deshielding can be distributed over many components and so become difficult to fully identify. However, noting the above and using the ADF NMR analysis package enables identification of the principal components that contribute to the $\sigma^p$ term of 1\* (Fig. 4) as being magnetic field-induced coupling between U≡N σ↔π\* and π↔σ\* MOs; these MO combinations conform to the requirement for rotational orthogonality and also being spatially and energetically proximate.

Field-induced magnetic mixing of the ground state with low-lying, thermally inaccessible, paramagnetic states in 1\*—i.e. temperature-independent paramagnetism, TIP—is confirmed

**Table 1 Natural localised molecular orbital (NLMO) contributions to the principal $^{15}N$ nuclear shielding components ($\sigma^d + [\sigma^p + \sigma^{so}]$) of 1\*.**

| NLMO[a] | SR | | | SOR | | | $\Delta^{sod}$ | | | %NBO | Occ. |
|---|---|---|---|---|---|---|---|---|---|---|---|
| | L[b] | NL[c] | L+NL | L[b] | NL[c] | L+NL | L[b] | NL[c] | L+NL | | |
| **$\sigma$-U $\equiv$ N** | | | | | | | | | | 98.7 | 1.98 |
| $\sigma_{iso}$ | −842 | −4 | −846 | −818 | −6 | −824 | 24 | −2 | 22 | | |
| $\sigma_{xx}$ | −1270 | −6 | −1276 | −1225 | −9 | −1234 | 45 | −3 | 42 | | |
| $\sigma_{yy}$ | −1270 | −6 | −1276 | −1225 | −8 | −1233 | 45 | −2 | 43 | | |
| $\sigma_{zz}$ | 13 | 0 | 13 | −2 | −1 | −3 | −15 | −3 | −16 | | |
| **$\pi$-U $\equiv$ N** | | | | | | | | | | 98.9 | 1.98 |
| $\sigma_{iso}$ | −307 | 16 | −291 | −319 | 16 | −303 | −12 | 0 | −12 | | |
| $\sigma_{xx}$ | −926 | 36 | −890 | −876 | 34 | −842 | 50 | −2 | 48 | | |
| $\sigma_{yy}$ | 0 | 0 | 0 | −78 | 3 | −75 | −78 | 3 | −75 | | |
| $\sigma_{zz}$ | 6 | 12 | 18 | −4 | 13 | 9 | −10 | 1 | −9 | | |
| **$\pi$-U $\equiv$ N** | | | | | | | | | | 98.7 | 1.98 |
| $\sigma_{iso}$ | −306 | 16 | −290 | −318 | 16 | −302 | −12 | 0 | −12 | | |
| $\sigma_{xx}$ | 0 | 0 | 0 | −79 | 2 | −77 | −79 | 2 | − 77 | | |
| $\sigma_{yy}$ | −923 | 36 | −887 | −873 | 35 | −838 | 50 | −1 | 49 | | |
| $\sigma_{zz}$ | 7 | 12 | 19 | −2 | 13 | 11 | −9 | 1 | −8 | | |
| **$1s_{core}$-N** | | | | | | | | | | 100 | 2.00 |
| $\sigma_{iso}$ | 240 | 0 | 240 | 240 | 0 | 240 | 0 | 0 | 0 | | |
| $\sigma_{xx}$ | 240 | 0 | 240 | 240 | 0 | 240 | 0 | 0 | 0 | | |
| $\sigma_{yy}$ | 240 | 0 | 240 | 239 | 0 | 239 | −1 | 0 | −1 | | |
| $\sigma_{zz}$ | 240 | 0 | 240 | 241 | 0 | 241 | 1 | 0 | 1 | | |
| **$\Sigma_{core}$-U** | | | | | | | | | | e | f |
| $\sigma_{iso}$ | −48 | 59 | 11 | −50 | 57 | 7 | −2 | −2 | −4 | | |
| $\sigma_{xx}$ | −72 | 89 | 17 | −76 | 77 | 1 | −4 | −12 | −16 | | |
| $\sigma_{yy}$ | −71 | 87 | 16 | −74 | 92 | 18 | −3 | 5 | 2 | | |
| $\sigma_{zz}$ | 0 | 0 | 0 | −1 | 2 | 1 | −1 | 2 | −1 | | |
| **$\Sigma_{other}$** | | | | | | | | | | e | f |
| $\sigma_{iso}$ | −58 | 42 | −14 | −42 | 37 | −5 | 16 | −5 | 9 | | |

[a]B3LYP calculations, all shielding parameters are in ppm. [b]Lewis contribution of the NLMO. [c]Non-Lewis contribution of the NLMO. [d]Defined as the SOR ($\sigma^d + [\sigma^p + \sigma^{so}]$) − SR ($\sigma^d + [\sigma^p + \sigma^{so}]$) to isolate the SO component. [e]Multiple NLMOs, but %NBOs all >85%. [f]Multiple NLMOs, but all occupancies >1.72 electrons per NLMO.

experimentally by SQUID magnetometry measurements on powdered 1\* (Fig. 5 and Supplementary Figs. 9–11), which reveal a small ($\chi_M T < 0.04$ cm$^3$ mol$^{-1}$ K) magnetic moment, but a shallow positive slope for the temperature-dependent $\chi_M T$ data even though $^1S_0$ uranium(VI) is formally a $5f^0 6d^0$ ion. Linear regression analysis reveals a $\chi_{TIP}$ value of $0.9527 \times 10^{-4}$ cm$^3$ mol$^{-1}$ K for 1\*, which compares to analogous values of $3.43 \times 10^{-4}$ and $6.26 \times 10^{-4}$ cm$^3$ mol$^{-1}$ K determined for two uranium(VI)-acetylides[24]. This suggests that TIP effects on the $\sigma^p$ term will be modest.

We note that the $\delta_{xx}$, $\delta_{yy}$, $\delta_{zz}$, and $\Omega$ values for N$_2$ are 136, 136, −539, and 675 ppm[59], which when compared to the data for 1\* reflects the presence of the Tren$^{TIPS}$ ligand in 1\* and, since the $\Delta\delta_{ppm}$ between 1\* and N$_2$ changes much less for $\delta_{zz}$ compared to $\delta_{xx}$ and $\delta_{yy}$, the expectation of strong mixing of occupied and virtual orbitals when a magnetic field is applied perpendicular to the U$\equiv$N bond[60–62]. Since the $\sigma^p$ term for 1\* is notably large, it follows that the U$\equiv$N bond is highly covalent, as has been proposed before on the basis of QTAIM calculations[39].

While the MO analysis provides a useful qualitatively instructive framework to probe the field-induced magnetic coupling of occupied and virtual orbitals, the delocalised nature of Kohn Sham MOs renders such analysis incomplete. In order to derive a more representative picture we analysed the shielding data in terms of NLMOs[63,64] (Table 1) pertaining to the NLMOs in Fig. 3. These data confirm that the $\delta_{iso}$ and $\Omega$ values of the nitride in 1\* is due primarily to the U$\equiv$N $\sigma$- and $\pi$-bonds, which induce large negative $\sigma_{xx}$ and $\sigma_{yy}$ nuclear shielding component contributions to the $\sigma^p$ parameter.

Focussing on the SOR data, the $\sigma$-bonding NLMO contributes −1234 and −1233 ppm to the Lewis and non-Lewis (L + NL) $\sigma_{xx}$

and $\sigma_{yy}$, of which −1262 and −1261 ppm are $\sigma^p + \sigma^{so}$, and 28 ppm of each are $\sigma^d$, respectively. Since the $\sigma_{zz}$ value is close to zero this then produces a $\sigma_{iso}$ of −824 ppm for the U$\equiv$N $\sigma$-bond NLMO. The two U$\equiv$N $\pi$-bonds present a slightly more complex picture, with L + NL $\sigma_{xx}$ and $\sigma_{yy}$ values of −870/−90 and −91/−866 for the $\sigma^p + \sigma^{so}$ component and $\sigma^d$ values of 28/15 and 14/28 ppm; these values reflect the strong response of a given $\pi$-orbital for a perpendicular magnetic field, but much weaker response for a parallel magnetic field, as shown most clearly by the SR data, but the SOR calculation resolves contributions resulting from the quasi-degeneracy, and thus symmetry breaking components, of the orthogonal $\sigma_{xx}$ and $\sigma_{yy}$ tensors that result from residing within the $C_3$-symmetric ligand field environment of the Tren$^{TIPS}$ ligand. Again, $\sigma_{zz}$ components are quite small, resulting in $\sigma_{iso}$ values of −303 and −302 ppm for the two U$\equiv$N $\pi$-bonds, showing that overall although the two U$\equiv$N $\pi$-bonds are not equivalent, their slight non-degeneracy averages out at the $\sigma_{iso}$ level.

The large U$\equiv$N $\sigma_{iso}$ NLMO values are counter-balanced largely only by the $\sigma^d$ value from the nitride 1$s$ orbital of 240 ppm, since various contributions from uranium outer core orbitals and other minor contributions from N lone pairs tend to be mainly cancelled out by competing Lewis and Non-Lewis components, leaving the U$\equiv$N $\sigma$- and $\pi$-bonding components as the dominant nuclear shielding components that primarily determine the NMR characteristics of the nitride.

A notable feature of the NLMO and MO shielding component data is the consistently small size of $\sigma^{so}$, whereas $\sigma^{so}$ of up to hundreds of ppm have been reported in NMR studies of uranium-carbon, -nitride, and chalcogenido bonding[20,21,24–26,36].

As shown consistently by NBO and NLMO analysis the N-component of the bonding of the nitride is dominated by $2p$-orbital character, with only 8% $2s$-character in the $U \equiv N$ $\sigma$-bond. Transfer of SO-induced spin-polarisation to the NMR nucleus is mediated by the N $s$-orbital, and so with little $2s$ character a small Fermi contact will result, despite the large overall contributions of both U- and N-orbitals in the $U \equiv N$ bond. Thus, despite the significant SOC that would be anticipated from a covalent $U \equiv N$ linkage with significant $5f$-character, there is little $\sigma^{so}$ in $\mathbf{1}^*$. Though the $\Delta^{so}$ values for the $\sigma$- and $\pi$-$U \equiv N$ bonds are small (cf +22 and −12 (av.)) we note that their respective signs are in-line with SO-heavy-atom-light-atom effects[37], and again highlight the dominant contribution of $\sigma \leftrightarrow \pi^*$ over $\pi \leftrightarrow \sigma^*$ mixing to the $^{15}N_{nitride}$ $\delta_{iso}$ of $\mathbf{1}^*$, as consistently evidenced by the MO and NLMO analyses.

It is instructive to consider the DFT $\sigma^{so}/2s$-character NLMO data for the nitride (−29 ppm/8%), amide (−40 ppm/37% av.), and amine (−20 ppm/17%). Recalling the Mayer and DI data above, the nitride is bonded to uranium very covalently with high $5f$- but low $2s$-character; therefore, $\sigma^{so}$ translates to being intermediate of the three. In contrast, the amide has lower, but significant enough, covalent bonding to uranium but much greater $2s$-character to facilitate larger $\sigma^{so}$ than the nitride, whereas the amine has essentially double the $2s$-character of the nitride, but it engages in much more electrostatic bonding to uranium giving a lower $\sigma^{so}$ value than the nitride.

The observation of little $\sigma^{so}$ for $\mathbf{1}^*$ may at first seem contradictory to previous reports of large $\sigma^{so}$ contributions to uranium(VI) chemical bonding. However, we propose that $\mathbf{1}^*$ simply reflects an extreme case of highly covalent bonding combined with little $s$-character in the bonding of the NMR nucleus. Where other ionic linkages are concerned, there is either more $s$-character in the bonding of the NMR nucleus, giving greater Fermi contact that would increase $\sigma^{so}$, or there is likely heavy-atom heavy-atom shielding effects[37]. Thus, even though $\mathbf{1}^*$ has major $5f$-bonding contributions and very high covalency, dominant $2p$-character of the N bonding results in small $\sigma^{so}$—this may be a hallmark of such covalent bonding with low $s$-character bonded ligands, and conversely it may be a characteristic feature of more ionic bonds, but ones which have enough covalency and $s$-character to have large $\sigma^{so}$ that is increased by $5f$-bonding character[21–26,36].

**Correlating $^{15}N$ NMR spectroscopic chemical shift to metal-nitride bond order.** The MO and NLMO data consistently suggest essentially constant $\sigma^d$ and small $\sigma^{so}$ contributions but large $\sigma^p$ contributions to the $^{15}N_{nitride}$ $\delta_{iso}$ resonance of $\mathbf{1}^*$, which when taken together with the unremarkable HOMO-LUMO gap and small $\chi_{TIP}$ value of $\mathbf{1}^*$ are consistent with the $\sigma^p$ parameter being the decisive factor for the $^{15}N_{nitride}$ $\delta_{iso}$. Since $\sigma^p$ is a direct reporter of covalency, then plots of the experimental $^{15}N$ $\delta_{iso}$ vs computed bond order metrics should produce linear correlations across all formal M–N bond orders. For this condition to be valid, the HOMO-LUMO gaps, which constitute a proportionate representation of the extent of magnetic field-induced mixing of occupied and virtual orbitals that contribute to $\sigma_{iso}$, for complexes under consideration should all be similar. This is indeed the case, spanning a range of ≤2.8 eV and we note that the HOMO-LUMO gap of $\mathbf{1}^*$ is consistently close to the mean and median computed HOMO-LUMO gap over BP86, PBE0, and B3LYP functionals (Supplementary Table 5). Indeed, separately plotting $^{15}N$ $\delta_{iso}$ values for the three unique, correlated U–N bonds in $\mathbf{1}^*$ (experimental nitride $\delta_{iso}$ and extrapolated computed amide and amine $\delta_{iso\ values}$) and available experimental data for 16 literature complexes[25,26,46–51,65]—various M-$NR_2$, M = NR, and M ≡ N (terminal and capped) bonds involving Th, Ti, Zr, V, and Mo—against Mayer, Nalewajski-Mrozek (NM), Delocalisation Index (DI), and interatomic exchange-correlation energy ($V_{XC}$)[66] bond order metrics produces satisfactory linear correlated relationships with $R^2$ values ranging from 0.7240 ($\delta_{iso}$ vs $V_{XC}$) to 0.8621 ($\delta_{iso}$ vs Mayer) (Fig. 6 and Supplementary Figs. 12–14 and Supplementary Table 6).

Considering the wide range of metals, metal oxidation states, supporting ligands, bridging and terminal nitrides, formal complex charges, solvent variation (THF and benzene), and focus on $\delta_{iso}$ that makes no distinction between dependency on $\sigma^p$ or ($\sigma^p + \sigma^{so}$), the agreement of the four correlations is remarkably good. Furthermore, focussing on $\delta_{iso}$ vs Mayer (Fig. 6) if $[(N'')_3Th(\mu-N)Th(N'')_3][K(18C6)(THF)_2]$[25,26] is omitted from the comparison, which is reasonable since that complex alone in the series exhibits large $\sigma^{so}$ effects (79 ppm) that compromise the linear $\sigma^p$ response, $R^2$ becomes 0.905. These plots with associated $\delta_{iso}$-bond order equations thus present reasonable models for making $^{15}N$ NMR spectroscopic $\delta_{iso}$ and bond order predictions. These correlation plots all consistently suggest that the $U \equiv N$ bond in $\mathbf{1}^*$ is highly covalent, and indeed more

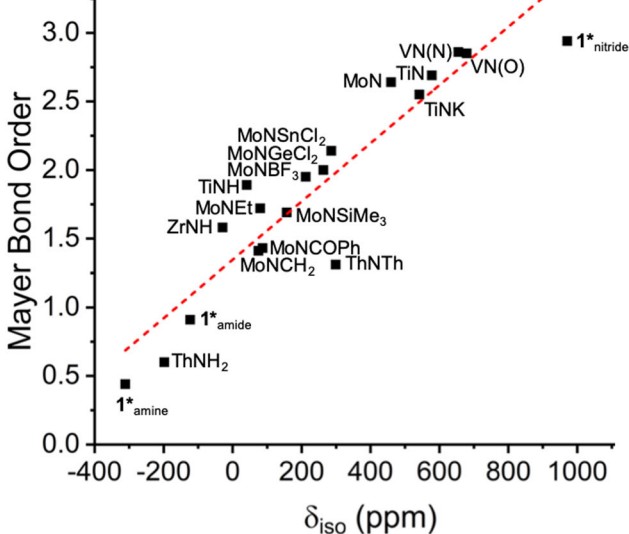

| Code | Formula |
|---|---|
| $\mathbf{1}^*$ | $[U(N^*)(Tren^{TIPS})]$ |
| VN(O) | $[V(N)(L^{MeDipp})(ODipp))]$ |
| VN(N) | $[V(N)(L^{MeDipp})\{N(Tol)(Mes)\}]$ |
| TiN | $[Ti(N)(NP)_2]^{1-}$ |
| TiNK | $[Ti(\mu-N)(NP)_2K(18C6)]$ |
| MoN | $[Mo(N)\{N(Bu^t)(C_6H_3-3,5-Me_2)\}_3]$ |
| MoNSnCl$_2$ | $[Mo(NSnCl_2)\{N(Bu^t)(C_6H_3-3,5-Me_2)\}_3]$ |
| MoNGeCl$_2$ | $[Mo(NGeCl_2)\{N(Bu^t)(C_6H_3-3,5-Me_2)\}_3]$ |
| MoNBF$_3$ | $[Mo(NBF_3)\{N(Bu^t)(C_6H_3-3,5-Me_2)\}_3]$ |
| MoNSiMe$_3$ | $[Mo(NSiMe_3)\{N(Bu^t)(C_6H_3-3,5-Me_2)\}_3]$ |
| MoNCOPh | $[Mo(NCOPh)\{N(Bu^t)(C_6H_3-3,5-Me_2)\}_3]$ |
| MoNEt | $[Mo(NEt)\{N(Bu^t)(C_6H_3-3,5-Me_2)\}_3]$ |
| MoNCH$_2$ | $[Mo(NCH_2)\{N(Bu^t)(C_6H_3-3,5-Me_2)\}_3]$ |
| TiNH | $[Ti(NH)(L^{ButDipp})(NTol_2)]$ |
| ZrNH | $[Zr(NH)(NP)_2]$ |
| ThNTh | $[(N'')_3Th(\mu-N)Th(N'')_3][K(18C6)(THF)_2]$ |
| ThNH$_2$ | $[Th(NH_2)(N'')_3]$ |

$Tren^{TIPS} = N(CH_2CH_2NSiPr^i_3)_3$; $L^{MeDipp} = HC(CMeNDipp)_2$; $L^{ButDipp} = HC(CBu^tNDipp)_2$; Dipp = 2,6-$Pr^i_2$-$C_6H_3$; Tol = 4-Me-$C_6H_4$; NP = MesNC$_6$H$_3$-3-Me-2-PPr$^i_2$; Mes = 2,4,6-Me$_3$-$C_6H_2$; N'' = N(SiMe$_3$)$_2$.

**Fig. 6 Correlation of experimental $^{15}N$ $\delta_{iso}$ data with computed Mayer bond orders.** Linear regression: Mayer bond order = (0.0019 × $\delta_{iso}$(exp)) + 1.3461, $R^2 = 0.8621$.

covalent than transition metal analogues, thus experimentally confirming the previous computational prediction[39].

## Discussion

We have reported solution and solid-state $^{15}N$ NMR spectroscopic characterisation of $\mathbf{1^*}$, revealing exceptional $^{15}N$ chemical shift and chemical shift anisotropy for N-containing compounds. These properties provide a benchmark for $^{15}N$ NMR spectroscopy, and result from large $\sigma^p$ values (and almost negligible $\sigma^{so}$ values) that originate from the $U \equiv N$ triple bond $\sigma$- and $\pi$-components. Thus, the $U \equiv N$ triple bond in $\mathbf{1^*}$ is experimentally confirmed to be highly covalent, and one that by computational and now experimental measures is more covalent than group 4–6 transition metal nitrides. The small $\sigma^{so}$ contributions reflect the low 2s-character of the nitride bonding orbitals, which reduces the expected large SOC from a highly covalent $U \equiv N$ triple bond with significant 5f-bonding character in $\mathbf{1^*}$. The data presented here have been combined with literature data to confirm correlated linear relationships between $^{15}N$ $\delta_{iso}$ data for $M \leftarrow NR_3$, $M-NR_2$, $M = NR$, and $M \equiv N$ linkages and four measures of bond order in good agreement, which can be used for $^{15}N$ NMR spectroscopic $\delta_{iso}$ and bond order predictions. By providing experimental confirmation of a previous theoretical prediction, these results demonstrate the powerful nature of using NMR spectroscopy to directly address the central, fundamental goal in actinide science of quantifying the nature and extent of covalency in An–L chemical bonding. These experimental data also serve to highlight the predictive power of modern 5f quantum chemistry.

## Methods

**Safety**. Depleted uranium (0.2% $^{235}U$, 99.8% $^{238}U$) is a weak α-emitter (4.197 MeV) with a half-life of $4.47 \times 10^9$ years. While most manipulations can be carried out safely using Schlenk line or glove boxes in an appropriate personal protective equipment, logging, and monitoring regime, particular care should be exercised when embarking on solid-state magic angle spinning (SS-MAS) NMR experiments. A full risk assessment and protocols for loading, unloading, and actions in the event of a rotor crash and contamination should be completed before undertaking SS-MAS-NMR experiments.

**General experimental details**. All manipulations were carried out under an inert atmosphere of dry nitrogen using Schlenk techniques, or an MBraun UniLab glovebox operating under an atmosphere of dry nitrogen. THF, toluene, and pentane solvents were dried by passage through activated alumina towers and degassed before use. Hexanes and benzene were distilled from potassium. All solvents were stored over potassium mirrors except for ethers which were stored over activated 4Å sieves. Deuterated solvents were distilled from potassium, degassed by three freeze–pump–thaw cycles and stored under nitrogen prior to use. Iodine was used as purchased. Sodium-1-$^{15}N$ azide (98%) and benzo-15-crown-5 (B15C5) were dried under vacuum for 8 h before use. $KC_8$[67] and [U(Tren$^{TIPS}$)(Cl)][38] were prepared using literature methods.

Solution $^1H$ and $^{29}Si\{^1H\}$ spectra were recorded on a Bruker AVIII 400 spectrometer operating at 400 and 79 MHz, respectively; chemical shifts are quoted in ppm and are relative to TMS ($^1H$, $^{29}Si$), respectively. Solution $^{15}N$ NMR spectra were acquired using a Bruker AVIII HD 400 spectrometer with 5 mm Prodigy probe. ($^1H$ TMS frequency 400 MHz, Me$^{15}NO_2$ frequency 40.5541637 MHz). Spectra were referenced to an external MeNO$_2$ (neat) standard and acquired both unlocked without adjusting the field correction relative to the standard (to ensure accurate chemical shift measurement), and locked. Acquisition was performed by averaging 24576 transients of 16384 complex points acquired using a 10° r.f. pulse with an acquisition time of 1 s and a recycle delay of 1.7 s. For processing, a matched filter (3.2 Hz Lorentzian broadening) was applied to provide an optimum signal-to-noise ratio. Solid-state {$^1H$-}$^{15}N$ cross-polarisation (CP) NMR spectra were recorded using a Bruker 9.4 T (400 MHz $^1H$ Larmor frequency) AVANCE III spectrometer equipped with a 4 mm HFX MAS probe. Experiments were acquired at ambient temperature using both static and MAS conditions. Samples were packed into 4 mm o.d. zirconia rotors in a glovebox, and sealed with a Kel-F rotor cap. The $^1H$ π-pulse duration was 5 μs, and the $^{15}N$ π-pulse duration was 40 μs for static experiments and 22 μs for MAS experiments. $^{15}N$ spin-locking was applied for 5 ms at ~12.5 kHz, with corresponding $^1H$ spin-locking at ~12.5 kHz, for static CP experiments and at ~23 kHz, with corresponding ramped (70–100%) $^1H$ spin-locking, for CPMAS experiments. One hundred kilohertz of SPINAL-64 (ref. [68]) heteronuclear $^1H$ decoupling was used throughout ~6 ms of signal acquisition (with 6.1 and 2.1 μs dwell-time between complex data points for

static and MAS experiments, respectively). A Hahn-echo $\tau_r$–π–$\tau_r$ sequence of two rotor periods total duration was applied to $^{15}N$ after CP to circumvent receiver dead-time. For static and fast MAS experiments ($\nu_r = 9$ kHz), the transmitter frequency offset was 1420 ppm and 912,576 and 204,800 transients were co-added for static and fast MAS experiments, respectively, with repetition delays of 0.6 s. For slow MAS experiments ($\nu_r = 2.5$ kHz), transmitter frequency offsets of 1820, 1420, 1020, and −380 ppm were used and the resulting magnitude spectra were added. Spectral simulations were performed in the solid line-shape analysis (SOLA) module v2.2.4 in Bruker TopSpin v4.0.9. The $^{15}N$ chemical shifts were referenced to MeNO$_2$ externally using glycine (−347.2 ppm)[69]. Static variable-temperature magnetic moment data were recorded in an applied dc field of 1 T on a Quantum Design MPMS 3 superconducting quantum interference device (SQUID) magnetometer using recrystallised powdered samples. Care was taken to ensure complete thermalisation of the sample before each data point was measured and samples were immobilised in an eicosane matrix to prevent sample reorientation during measurements. Diamagnetic corrections were applied using tabulated Pascal constants and measurements were corrected for the effect of the blank sample holders (flame sealed Wilmad NMR tube and straw) and eicosane matrix.

**Preparation of [U(Tren$^{TIPS}$)($^{15/14}N^{14}N^{14/15}N$)]**. This synthesis was performed in an analogous manner to the preparation of the previously reported [U(Tren$^{TIPS}$)(N$_3$)][41], using 98% isotopically enriched sodium-1-$^{15}N$ azide in place of isotopically normal sodium azide. THF (40 ml) was added slowly to a pre-cooled stirring mixture of [U(Tren$^{TIPS}$)(Cl)] (1.77 g, 2.00 mmol) and sodium-1-$^{15}N$ azide (0.20 g, 3.00 mmol) at −78 °C. After addition, the brown solution was allowed to warm to room temperature and stirred for 3 days to ensure all the uranium chloride starting material was fully converted to the uranium azide product, which was confirmed by $^1H$ NMR of the reaction before following work-up. Volatiles were removed in vacuo and the product was extracted into toluene (30 ml). The mixture was filtered, and volatiles were removed in vacuo to give a brown crude product, which was washed with cold pentane (2 × 10 ml) to yield [U(Tren$^{TIPS}$)($^{15/14}N^{14}N^{14/15}N$)] as a green solid. Yield: 1.34 g, 75%. The purity of [U(Tren$^{TIPS}$)($^{15/14}N^{14}N^{14/15}N$)] was confirmed by $^1H$ NMR spectroscopy, evidenced by an identical spectrum to that of the non-labelled analogue[39].

**Preparation of [U(Tren$^{TIPS}$)(N*)][K(B15C5)$_2$] (N* = 50:50 $^{14/15}N$)**. This synthesis was carried out in an analogous manner to the preparation of the reported [U(Tren$^{TIPS}$)(N)][K(B15C5)$_2$][41,42], using [U(Tren$^{TIPS}$)($^{15/14}N^{14}N^{14/15}N$)] instead of non-$^{15}N$-labelled [U(Tren$^{TIPS}$)(N$_3$)][39]. Toluene (30 ml) was added to a pre-cooled stirring mixture of [U(Tren$^{TIPS}$)($^{15/14}N^{14}N^{14/15}N$)] (1.34 g, 1.50 mmol) and KC$_8$ (0.20 g, 1.50 mmol) at −78 °C, and then the resulting mixture was allowed to warm to ambient temperature and stirred for further 6 days to afford a dark brown suspension. After this time, volatiles were removed in vacuo and the product was washed with pentane (3 × 10 ml). The dark brown solid residue was then extracted into hot benzene (80 °C) and quickly filtered through a frit to remove the graphite precipitate. The filtrate was stored at 5 °C for 24 h to yield dark red crystals of [{U(Tren$^{TIPS}$)}$_2$(μ-N*K)$_2$] that were isolated by filtration and dried in vacuo. Yield: 0.57 g, 42% (crystalline). It should be noted that the isolated [{U(Tren$^{TIPS}$)}$_2$(μ-N*K)$_2$] and the following complexes [U(Tren$^{TIPS}$)(N*)][(B15C5)$_2$] and $\mathbf{1^*}$ are only 50% $^{15}N$-labelled due to the loss of 50% $^{15}N$-labelled nitrogen as dinitrogen in this reduction step. The bridging complex [{U(Tren$^{TIPS}$)}$_2$(μ-N*K)$_2$] was converted to the terminal species [U(Tren$^{TIPS}$)(N*)][(B15C5)$_2$] by further reacting with 2.0 equivalents of benzo-15-crown-5 ether; therefore, toluene (20 ml) was added to the obtained [{U(Tren$^{TIPS}$)}$_2$(μ-N*K)$_2$] (0.57 g, 0.63 mmol) with benzo-15-crown-5 (B15C5, 0.34 g, 1.30 mmol) at −78 °C. The red brown mixture was allowed to warm to room temperature and stirred for 16 h. The solvent was removed in vacuo to yield a brown residue which was washed with pentane (2 × 10 ml) and dried in vacuo to yield [U(Tren$^{TIPS}$)(N*)][(B15C5)$_2$] as a brown powder. Yield: 0.58 g, 64%. The purity of [U(Tren$^{TIPS}$)(N*)][(B15C5)$_2$]was confirmed by $^1H$ NMR spectroscopy, evidenced by an identical spectrum to that of the non-labelled analogue[42].

**Preparation of [U(Tren$^{TIPS}$)(N*)] (1*, N* = 50:50 $^{14/15}N$)**. The synthesis of $\mathbf{1^*}$ was carried out in an analogous manner to the reported preparation of $\mathbf{1}$ (ref. [39]), using [U(Tren$^{TIPS}$)(N*)][(B15C5)$_2$] as the reacting precursor in place of non-$^{15}N$-labelled normal uranium(V) nitride complex [U(Tren$^{TIPS}$)(N)][(B15C5)$_2$][42]. A solution of I$_2$ (0.05 g, 0.20 mmol) in toluene (10 ml) was added dropwise to a stirring solution of [U(Tren$^{TIPS}$)(N*)][(B15C5)$_2$] (0.58 g, 0.40 mmol) in toluene (10 ml) at −78 °C. The brown solution was allowed to warm to room temperature with stirring over 16 h under dark as $\mathbf{1^*}$ is light sensitive. After this time, volatiles were removed in vacuo and the product was washed with pentane (2 × 5 ml). The brown residue was extracted into toluene (10 ml) and filtered to remove the [K(B15C5)$_2$]I salt. The resulting red filtrate was concentrated to 3 ml and stored at −35 °C for 2 days, yielding $\mathbf{1^*}$ as red crystals. Yield: 0.18 g, 51%. The purity of the product was confirmed by $^1H$ NMR in C$_6$D$_6$, evidenced by an identical spectrum to that of the non-labelled analgoue[39]. Complex $\mathbf{1^*}$ is poorly soluble in pentane, and only partially soluble in benzene/toluene, but more soluble in THF, so the solution $^{15}N$ NMR spectrum was obtained in D$_8$-THF, the $^1H$ NMR of $\mathbf{1^*}$ was also recorded in D$_8$-THF, indicating good stability of $\mathbf{1^*}$ in THF solvent in the absence of ambient light. $^1H$ NMR (400 MHz, D$_8$-THF, 298 K): δ (ppm) 1.56 (d, 54H,

$CH(CH_3)_2$), 2.20 (septet, 9H, $CH(CH_3)_2$), 3.04 (t, 6H, $CH_2CH_2$), 5.42 (t, 6H, $CH_2CH_2$). $^1$H NMR (400 MHz, $D_8$-THF and $C_6D_6$ = 50:50, 298 K): δ (ppm) 1.54 (d, 54H, $CH(CH_3)_2$), 2.16 (septet, 9H, $CH(CH_3)_2$), 2.68 (t, 6H, $CH_2CH_2$), 5.20 (t, 6H, $CH_2CH_2$). $^1$H NMR (400 MHz, $C_6D_6$, 298 K): δ (ppm) 1.69 (d, 54H, $CH(CH_3)_2$), 2.31 (septet, 9H, $CH(CH_3)_2$), 2.48 (t, 6H, $CH_2CH_2$), 5.17 (t, 6H, $CH_2CH_2$). $^{29}$Si{$^1$H} NMR (79 MHz, $D_8$-THF, 298 K): δ (ppm) 3.78. $^{29}$Si{$^1$H} NMR (79 MHz, $D_8$-THF and $C_6D_6$ = 50:50, 298 K): δ (ppm) 5.60. $^{29}$Si{$^1$H} NMR (79 MHz, $C_6D_6$, 298 K): δ (ppm) 5.87. $^{15}$N{$^1$H} NMR (41 MHz, $D_8$-THF, 298 K): δ (ppm) 968.9. $^{15}$N{$^1$H} NMR (41 MHz, $D_8$-THF and $C_6D_6$ = 50:50, 298 K): δ (ppm) 970.4. $^{15}$N{$^1$H} NMR (41 MHz, $C_6D_6$, 298 K): δ (ppm) 972.6.

**General computational details.** Geometry optimisations were performed using coordinates derived from the respective crystal structures as the starting points. No constraints were imposed on the structures during the geometry optimisations. The calculations were performed using the Amsterdam Density Functional (ADF) suite version 2017 with standard convergence criteria[70,71]. The DFT geometry optimisations employed Slater type orbital (STO) TZP polarisation all-electron basis sets (from the Dirac and ZORA/TZP database of the ADF suite). Scalar relativistic approaches (spin–orbit neglected) were used within the ZORA Hamiltonian[72–74] for the inclusion of relativistic effects and the local density approximation with the correlation potential due to Vosko et al. was used in all of the calculations[75]. Generalised gradient approximation corrections were performed using the functionals of Becke and Perdew[76,77].

Scalar and two-component spin–orbit relativistic (ZORA) single point energy calculations were then run on the geometry optimised coordinates. For **1**\* the functionals screened included SAOP-TZ2P, BP86-TZ2P, PBE0-TZP, PBE0-TZ2P, PBE0-HF40-TZ2P, and B3LYP-TZ2P, the latter of which gave the closest agreement of computed NMR properties compared to experiment. Once B3LYP was identified as the functional of choice, scalar relativistic (ZORA) single point energy calculations on the literature compounds used for the standardisation of empirical corrections were run with the B3LYP functional and STO TZ2P polarisation all-electron basis sets. The conductor-like screening model (COSMO) was used to simulate solvent effects, with the appropriate solvent matched to the solvent medium used in the literature report for the specific compound being modelled. MOs were visualised with ADFView.

NLMO analysis of **1**\* was carried out using NBO6 (ref. [78]) and the B3LYP STO TZ2P scalar relativistic ZORA COSMO single point energy data. These calculations used the Hartree–Fock RI scheme to suspend the dependency key and avoid numerical issues. The NLMOs were visualised using ADFView.

Mayer and Nalewajski-Mrozek values were computed using ADF and the PBE0 functional. DI and $V_{XC}$ bond metrics were computed using AIMAll[79], from WFX files generated from single point energy calculations performed with Gaussian 16 (ref. [80]). In these calculations, the PBE0 (refs. [81,82]) density functional approximation was employed. Dunning's correlation consistent basis sets of polarised triple-ζ quality[83–86] was used for all non-actinide atoms, except for K which was treated with Pople's 6-311G\* basis set[87]. U and Th atoms were treated with the all-electron SARC basis sets[88–90], including the second-order Douglas–Kroll–Hess (DKH2) Hamiltonian to account for scalar relativistic effects[91–93]. The PBE0 functional was used since NBO and NLMO outputs from the ADF and Gaussian 16 programmes could be checked to ensure consistency of outputs.

NMR shielding calculations were carried out using the NMR programme within ADF[63,64,94–98]. Calculated nuclear shieldings were converted to chemical shifts by subtraction from the calculated nuclear shielding of $MeNO_2$ in neat $MeNO_2$ calculated at the same level. Canonical MO contributions to the nuclear shieldings were calculated at the scalar and two-component spin–orbit levels, the former with the FAKESO key. Scalar and two-component spin–orbit NLMO calculations of the computed nuclear shieldings were carried out using NBO6 and ADF. These calculations used the Hartree–Fock RI scheme to suspend the dependency key and avoid numerical issues.

## Data availability
All data are available within this article, the Supplementary Information (Supplementary Figs. 1–14 and Table 1–6), or from S.T.L. on reasonable request.

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

## Acknowledgements

We gratefully acknowledge funding and support from the UK Engineering and Physical Sciences Research Council (grants EP/K024000/1, EP/M027015/1, and EP/S033181/1), European Research Council (grant CoG612724), and The University of Manchester including computational resources and associated support services from the Computational Shared Facility. The Alexander von Humboldt Foundation is thanked for a Friedrich Wilhelm Bessel Research Award to S.T.L. We thank Dr. Ashley J. Wooles (The University of Manchester) for MAS NMR sample radiological safety assessments and protocols.

## Author contributions

J.D. prepared and characterised the sample. J.A.S. performed and analysed the magnetic measurements. V.E.J.B. and N.K. conducted quantum chemical calculations to determine the bond metrics. R.W.A. and D.L. recorded and analysed the NMR spectroscopic data. S.T.L. conceived the central idea, conducted the DFT and NLMO NMR calculations, analysed the data, and wrote the manuscript with contributions from all authors.

## Competing interests

The authors declare no competing interests.
