## [Peer Review File · Nature Communications]

Exceptional Uranium(VI)-Nitride Triple Bond Covalency from ^{15}N Nuclear Magnetic Resonance Spectroscopy and Quantum Chemical AnalysisREVIEWER COMMENTS

Reviewer #1 (Remarks to the Author):

The manuscript by Liddle et. al. provides convincing arguments that their ^{15}N labeled $\text{U}\equiv\text{N}$ complex (1^*) contains a highly covalent $\text{U(VI)} - \text{N}$ bond, even more so than transition metal analogous. Their study makes extensive use of NMR (SS-NMR as well) revealing an extreme ^{15}N chemical shift and chemical shift anisotropy, which are the largest reported for any N-containing compound to date, as well as computational analyses which had to be fine-tuned for agreement. All of their NMR and computational analyses are sound and their comparisons to other known nitride containing complexes (both transition metals and actinides) in a variety of different oxidation states provides nice context.

The points below are raised for clarity and/or contextual purposes.

- 1.) It would be helpful if there was a comparison of the known actinide-nitride bond distances as determined from single crystal x-ray diffraction.
- 2.) In Figure 2, the σ -bonding orbital appears to be primarily composed of a 5d-orbital rather than a 5f orbital as suggested by computational analysis.

This study is sure to be of interest to those interested in NMR studies, actinide covalency, chemical bonding, and computational analyses. Thus, this study fits the goals of the journal and is worthy of acceptance.

Reviewer #2 (Remarks to the Author):

Review of "Exceptional Uranium(VI)-Nitride Triple Bond Covalency Measured by ^{15}N Nuclear Magnetic Resonance Spectroscopy"

Synopsis: The manuscript reports the ^{15}N NMR spectra of a uranium terminal nitride complex in solution and the solid state. The authors then calculate the isotropic ^{15}N chemical shift of the nitride using DFT with several different functionals. The functionals give widely varying values for the chemical shift. The authors select the functional with the best agreement with isotropic chemical shift, B3LYP, to perform further analysis of the ^{15}N NMR spectrum. The calculated bonding and NMR properties of the terminal nitride complex are then reported for the complex. Finally, the calculated bond-orders for several complexes are correlated with their isotropic ^{15}N chemical shifts.

Recommendation: Major revision and publish elsewhere. The bonding in these complexes has already been studied computationally, which showed that the uranium nitride interaction is highly covalent (ref 39). While the ^{15}N NMR results are impressive, they are not likely to be of interest to a wide audience. The manuscript should be published in a more narrowly focused journal.

General comments: The experimental work in this manuscript is performed well and described well. In particular, the fact that the authors were able to determine the chemical shift tensor for the terminal nitride is impressive and commendable. The computational work itself is performed well and described well. However, the analysis and explanation of these results is extremely confusing. Furthermore, the interpretation of the results appears to be incorrect. The authors imply that the isotropic chemical shift in this compound is only a function of the bond order (Figure 5). This is not accurate. The nature of the chemical bonding involving the atom is only one of three factors that significantly affect the chemical shift.

Specific comments:

- 1) The title of this manuscript is misleading. The covalency has not been measured by ^{15}N NMR spectroscopy. The shielding tensors were measured and compared with calculated shielding tensors. The covalency in the complex was calculated, not measured. A more appropriate wording would be,

“Exceptional Uranium(VI)-Nitride Triple Bond Covalency Implied by ^{15}N Nuclear Magnetic Resonance Spectroscopy.” However, for reasons explained below, even this seems inaccurate.

2) Please include the formulas for the different components of the chemical shift (Ramsey's formulas) in the text. Part of the reason that the paper is so confusing is that it refers to the diamagnetic, paramagnetic, and spin-orbit (SO) contributions to the chemical shift without ever defining them or providing the formulas for the reader.

3) The manuscript assumes that the degree of U-N covalency is the only important factor in determining the ^{15}N chemical shift. This assumption underlies the use of Figure 5 to correlate the ^{15}N chemical shift with the Mayer bond order. Unfortunately, this assumption is not valid. While the ^{15}N paramagnetic chemical shift should be approximately linearly related to the degree of covalency as noted by the authors, the chemical shift is also inversely proportional to the energy of the excited state responsible for the chemical shift. This latter contribution has not been adequately addressed in the manuscript. In particular, the excited states are relatively low in energy in U(VI) complexes, as shown by their relatively large temperature independent paramagnetism. As a result, the ^{15}N chemical shifts of U(VI) complexes will be larger than expected based on covalency alone. This is presumably the reason that the data point corresponding to the ^{15}N chemical shift of the uranium nitride in Figure 5 deviates so much from the predicted value.

4) How exactly is the SO contribution to the chemical shift calculated in ADF? The reported values are lower than expected for a light atom bonded to uranium.

5) The following explanation for the small SO contribution to the nitride chemical shift is incorrect: “Where more ionic linkages are concerned, sso is quenched less, and in many cases there is also more 2s-character in the heteroatom bonding than for the nitride, which would increase sso by Fermi contact.” Strongly covalent bonding will weaken the impact of SO coupling on the uranium center, not the SO contribution to the nitrogen chemical shift. Strongly covalent bonding will increase the contribution of SO to the nitrogen chemical shift. The cause of the small contribution of SO to the chemical shift could be that the N 2s orbital of the nitride is not strongly involved in bonding (the N 2p orbitals are close in energy to the U 5f orbitals and dominate bonding). The SO contribution to the chemical shift is mediated by the N 2s character in the bond, which only contains ~ 1% N 2s character.

6) Figure 3b appears to be missing two excited states that contribute strongly to the ^{15}N chemical shift. The occupied π -bonding states have $J_z = \pm 1$. These will be coupled to excited states with an unpaired electron in an f-orbital with $J_z = 0$ (shown) and $J_z = \pm 2$ (not shown). Did ADF not identify these as states that contributed to the paramagnetic component of the chemical shift?

Reviewer #3 (Remarks to the Author):

Contrary to the manuscript submitted by the authors this reviewer report is going to be somewhat boring, my apologies.

The authors ought to be congratulated for this work. It is exactly the kind of paper I would like to read. It concentrates on a relatively small topic and thoroughly analysis it. Deeper understanding is gained. The topic is certainly interesting enough to merit publication in Nature Communications and so is the data presented.

The theoretical analysis is exceedingly clear. I completely agree with the authors reasoning for the small values of SOC contributions. The included pictures and tables are chosen well and help to understand the presented results.

The cited literature seems to me to be complete and the SI contains all additional information one could wish for. I would like to commend the authors for including the xyz coordinates of the optimised structures. This should be the standard, but alas it is not.

I have been thinking hard on something to criticise in the submission, but I have given up.

My recommendation is therefore to publish the submitted paper as it is in Nature Communications.

We thank the referees for their comments.

Below we have appended the referee comments as received and provide our responses on a point-by-point basis. We believe that our responses and revisions ameliorate the concerns raised.

REVIEWER COMMENTS

Reviewer #1 (Remarks to the Author):

The manuscript by Liddle et. al. provides convincing arguments that their ^{15}N labeled $\text{U}\equiv\text{N}$ complex (**1***) contains a highly covalent $\text{U(VI)} - \text{N}$ bond, even more so than transition metal analogous. Their study makes extensive use of NMR (SS-NMR as well) revealing an extreme ^{15}N chemical shift and chemical shift anisotropy, which are the largest reported for any N-containing compound to date, as well as computational analyses which had to be fine-tuned for agreement. All of their NMR and computational analyses are sound and their comparisons to other known nitride containing complexes (both transition metals and actinides) in a variety of different oxidation states provides nice context.

RESPONSE: We thank the referee for appreciating our work.

The points below are raised for clarity and/or contextual purposes.

1.) It would be helpful if there was a comparison of the known actinide-nitride bond distances as determined from single crystal x-ray diffraction.

RESPONSE: We agree that a list of AnN distances would be of use if the corresponding ^{15}N δ_{iso} data were also available, however that is not the case, so a list of bond lengths would in our view on balance not add meaningful insight to the present discussion and could well become a distraction. However, we are all for providing context when appropriate so in the computational section we now when discussing the UN distance in **1*** give the corresponding distances for $[\text{U}^{\text{V}}(\text{N})(\text{Tren}^{\text{TIPS}})][\text{Na}(12\text{C}4)_2]$ and $[\text{U}^{\text{VI}}(\text{N})\{\text{OSi}(\text{OBu}^{\text{t}})_3\}_4][\text{NBu}^{\text{n}}_4]$ to provide some context to readers – they are the only other terminal uranium-nitrides for comparison.

2.) In Figure 2, the σ -bonding orbital appears to be primarily composed of a 5d-orbital rather than a 5f orbital as suggested by computational analysis.

RESPONSE: We thank the referee for prompting us to check, and can confirm that the UN NLMO σ -bond as depicted in figure 2 is indeed of predominantly 5f character. We noticed a small typo, that in the breakdown 5d should have been 6d which has been corrected. This orbital is essentially a hybrid of fz3 and dz2 components, and when considering how they will hybridise, recalling that d is *gerade* and f *ungerade* symmetry, it can be readily seen that putting fz3 and dz2 orbitals side-by-side and arbitrarily setting the wavefunction signs to top lobe +, top toroid -, bottom toroid +, bottom lobe -, and top lobe +, toroid -, bottom lobe + that the combined top lobes will enjoy constructive overlap, the bottom lobes will experience destructive overlap, and the 2+1 toroid mismatch will be constructive for the top toroid but destructive for the lower toroid, so the overall result is that depicted in figure 2 – the destructive overlap lobe is still there, but since greatly diminished it cannot be seen at that angle due to the lower toroid obscuring it. These results have been checked with NBO analysis on Gaussian and ADF outputs giving almost identical results.

This study is sure to be of interest to those interested in NMR studies, actinide covalency, chemical bonding, and computational analyses. Thus, this study fits the goals of the journal and is worthy of acceptance.

RESPONSE: We thank the referee for their considered comments and support.

Reviewer #2 (Remarks to the Author):

Review of “Exceptional Uranium(VI)-Nitride Triple Bond Covalency Measured by ^{15}N Nuclear Magnetic Resonance Spectroscopy”

Synopsis: The manuscript reports the ^{15}N NMR spectra of a uranium terminal nitride complex in solution and the solid state. The authors then calculate the isotropic ^{15}N chemical shift of the nitride using DFT with several different functionals. The functionals give widely varying values for the chemical shift. The authors select the functional with the best agreement with isotropic chemical shift, B3LYP, to perform further analysis of the ^{15}N NMR spectrum. The calculated bonding and NMR properties of the terminal nitride complex are then reported for the complex. Finally, the calculated bond-orders for several complexes are correlated with their isotropic ^{15}N chemical shifts.

RESPONSE: Picking the functional that has the best agreement with experiment, and then using it to perform further analysis, is standard practice in DFT in many areas of chemistry and physics. Indeed, when looking at NMR analysis papers in the literature this is the same approach adopted by all other groups.

Recommendation: Major revision and publish elsewhere. The bonding in these complexes has already been studied computationally, which showed that the uranium nitride interaction is highly covalent (ref 39). While the ^{15}N NMR results are impressive, they are not likely to be of interest to a wide audience. The manuscript should be published in a more narrowly focused journal.

RESPONSE: On a regular basis we receive reviews of our manuscripts where referees ask for all manner of computational additions because of their stated mistrust of the accuracy of DFT. It is therefore the case that whilst we think DFT is now quite reliable, many do not so whilst we have our prior computational analysis on its own it has always been the case that experimental proof was needed to convince the community fully of its veracity. This paper presents that experimental proof. Furthermore, as well as exhibiting record ^{15}N NMR properties that redefine ^{15}N NMR spectroscopy parameter space, the realisation of exceptional covalency is certainly surprising and we suggest that f covalency being greater than d is highly notable - it will surprise many - and is thus of interest to a broad readership. We also note that this journal recently published a paper on Ce-C bond covalency probed by ^{13}C NMR spectroscopy (*Nat. Commun.* **12**, 1713 (2021)), so there is precedent for interest in this area in this journal, and indeed RIXS studies on 5f covalency published in this journal (*Nat. Commun.* **8**, 16053 (2017)) point to a wider interest in f covalency by this journal.

General comments: The experimental work in this manuscript is performed well and described well. In

particular, the fact that the authors were able to determine the chemical shift tensor for the terminal nitride is impressive and commendable. The computational work itself is performed well and described well. However, the analysis and explanation of these results is extremely confusing. Furthermore, the interpretation of the results appears to be incorrect. The authors imply that the isotropic chemical shift in this compound is only a function of the bond order (Figure 5). This is not accurate. The nature of the chemical bonding involving the atom is only one of three factors that significantly affect the chemical shift.

RESPONSE: In the computational chemical shift analysis of **1*** section we referenced Ramsey's formula and explained that σ^p is inversely proportional to $1/r^3$ and the energy gap between ground and excited states that mix in a magnetic field, and the symmetry aspects. We felt it was evident that the σ^p component was largely reflecting the covalency, otherwise a linear correlation plot simply would not be obtained. However, we have provided further clarifying points as described below - these validate our argument.

Specific comments:

1) The title of this manuscript is misleading. The covalency has not been measured by ^{15}N NMR spectroscopy. The shielding tensors were measured and compared with calculated shielding tensors. The covalency in the complex was calculated, not measured. A more appropriate wording would be, "Exceptional Uranium(VI)-Nitride Triple Bond Covalency Implied by ^{15}N Nuclear Magnetic Resonance Spectroscopy." However, for reasons explained below, even this seems inaccurate.

RESPONSE: We agree that covalency has not been literally measured, rather it is proportional to the σ^p term that can be calculated. We explain below why σ^p is indeed proportional to covalency. Therefore we have changed the title to be "*Exceptional Uranium(VI)-Nitride Triple Bond Covalency from ^{15}N Nuclear Magnetic Resonance Spectroscopy and Quantum Chemical Analysis*".

2) Please include the formulas for the different components of the chemical shift (Ramsey's formulas) in the text. Part of the reason that the paper is so confusing is that it refers to the diamagnetic, paramagnetic, and spin-orbit (SO) contributions to the chemical shift without ever defining them or providing the formulas for the reader.

RESPONSE: We have now added three equations, the original Ramsey relationship of σ^d and σ^p to σ_{iso} , the version with σ^{so} added in, and a reduced version of the σ^p term. We have rewritten the associated paragraph to accommodate these changes and to add the definitions in. We then provide key references for the interested reader since a research paper is not the place to reproduce an extended description of the theory of NMR spectroscopy.

3) The manuscript assumes that the degree of U-N covalency is the only important factor in determining the ^{15}N chemical shift. This assumption underlies the use of Figure 5 to correlate the ^{15}N chemical shift with the Mayer bond order. Unfortunately, this assumption is not valid. While the ^{15}N paramagnetic chemical shift should be approximately linearly related to the degree of covalency as noted by the authors, the chemical is also inversely proportional to the energy of the excited state responsible for the chemical shift. This latter contribution has not been adequately addressed in the manuscript. In particular, the excited states are relatively low in energy in U(VI) complexes, as shown by their relatively large temperature independent paramagnetism. As a result, the ^{15}N chemical shifts of U(VI) complexes will be larger than expected based on covalency alone. This is presumably the reason that the data point corresponding to the ^{15}N chemical shift of the uranium nitride in Figure 5 deviates so much from the predicted value.

RESPONSE: Before submission we satisfied ourselves that the HOMO-LUMO gaps, which can be taken as a representative gauge to the efficiency of magnetically induced mixing, are quite similar across the series we have analysed, with the computed values being consistently in a relatively narrow range of ≤ 2.8 eV when computed with BP86, PBE0, and B3LYP functionals. Furthermore, across all 3 functionals the HOMO-LUMO gap of **1*** is consistently close to the mean and median HOMO-LUMO gaps for the series of 17 complexes. This removes the energy argument from considerations. We agree that TIP effects operate, but from linear regression we obtain a χ_{TIP} value of $0.9527 \times 10^{-4} \text{ cm}^3 \text{ mol}^{-1} \text{ K}$, which compares to analogous

values of 3.43×10^{-4} and $6.26 \times 10^{-4} \text{ cm}^3 \text{ mol}^{-1} \text{ K}$ determined for two uranium(VI)-acetylides (reference 24); **1*** being an order of magnitude smaller than those complexes suggests the TIP is in fact rather small. The only reasonable conclusion then is that the σ^p is the dominant term – if it were not we would not get anything close to a linear correlation, so the fact that we do confirms the validity of the position that the σ^p not only dominates but is proportional to the covalency (as is clear from Ramsey's formula and as a general principle also widely accepted as being the case). As we stated already, the linear correlation combines a great number of variables, yet still holds, so it is not a surprise that most complexes lie off the line, but more importantly they all handrail the line. To highlight these points to the reader, we have deposited a table of HOMO-LUMO gaps into the SI (Table 5) and at the appropriate points stated: (i) that the HOMO-LUMO gap of **1*** is unremarkable so it is not going to dominate the σ^p term so will be subordinate to the $1/r^3$ term; (ii) stated our χ_{TIP} and overtly compared it to the acetylide complexes mentioned above to show it is a small component; (iii) pointed out in the linear correlation section that the HOMO-LUMO gaps need to be similar for our premise to be valid, which they are so it is.

4) How exactly is the SO contribution to the chemical shift calculated in ADF? The reported values are lower than expected for a light atom bonded to uranium.

RESPONSE: ADF calculates the SO contribution as described in S.K. Wolff, T. Ziegler, E. van Lenthe, E.J. Baerends, *J. Chem. Phys.* **110**, 7689 (1999) and J. Autschbach, *Mol. Phys.* **111**, 2544 (2013). As we explained in the manuscript, the small SO contribution for the nitride is almost certainly due to the rather small 2s contribution to its bonding (8%) meaning there is little Fermi contact.

5) The following explanation for the small SO contribution to the nitride chemical shift is incorrect: "Where more ionic linkages are concerned, sso is quenched less, and in many cases there is also more 2s-character in the heteroatom bonding than for the nitride, which would increase sso by Fermi contact." Strongly covalent bonding will weaken the impact of SO coupling on the uranium center, not the SO contribution to the nitrogen chemical shift. Strongly covalent bonding will increase the contribution of SO to the nitrogen chemical shift. The cause of the small contribution of SO to the chemical shift could be that the N 2s orbital of the nitride is not strongly involved in bonding (the N 2p orbitals are close in energy to the U 5f orbitals and dominate bonding). The SO contribution to the chemical shift is mediated by the N 2s character in the bond, which only contains ~ 1% N 2s character.

RESPONSE: Re-reading our text we realise we were not clear enough and we agree with the referee, so have rewritten these parts and thank the referee for spotting this deficiency.

6) Figure 3b appears to be missing two excited states that contribute strongly to the ^{15}N chemical shift. The occupied π -bonding states have $J_z = \pm 1$. These will be coupled to excited states with an unpaired electron in an f-orbital with $J_z = 0$ (shown) and $J_z = \pm 2$ (not shown). Did ADF not identify these as states that contributed to the paramagnetic component of the chemical shift?

RESPONSE: The referee is correct, ADF did not identify these states as contributing to the paramagnetic component. As we (in the manuscript) and others (*e.g. Chem. Sci.* **8**, 1209 (2017); *Inorg. Chem.* **59**, 10138 (2020)) have noted, this approach does not always pick up all components, because a great number of components need to be deconvoluted which is challenging, and this is why we extended the analysis to include the NLMO-NMR approach since this does an excellent job of capturing the essential components.

Reviewer #3 (Remarks to the Author):

Contrary to the manuscript submitted by the authors this reviewer report is going to be somewhat boring, my apologies.

The authors ought to be congratulated for this work. It is exactly the kind of paper I would like to read. It concentrates on a relatively small topic and thoroughly analysis it. Deeper understanding is gained. The topic is certainly interesting enough to merit publication in Nature Communications and so is the data presented.

The theoretical analysis is exceedingly clear. I completely agree with the authors reasoning for the small values of SOC contributions. The included pictures and tables are chosen well and help to understand the presented results.

The cited literature seems to me to be complete and the SI contains all additional information one could wish for. I would like to commend the authors for including the xyz coordinates of the optimised structures. This should be the standard, but alas it is not.

I have been thinking hard on something to criticise in the submission, but I have given up.

My recommendation is therefore to publish the submitted paper as it is in Nature Communications.

RESPONSE: We thank the referee for their considered comments and support.

---End---

REVIEWER COMMENTS

Reviewer #1 (Remarks to the Author):

All of my concerns have been addressed in the revision.

Reviewer #2 (Remarks to the Author):

The authors have addressed my concerns. The manuscript should be published in Nature Communications.

I have a minor comment: the second sentence in the abstract should be rewritten, there is something wrong with the wording of the second clause. Should it be , "but this range sits between ionic lanthanide and ..."?

REVIEWER COMMENTS

Reviewer #1 (Remarks to the Author):

All of my concerns have been addressed in the revision.

RESPONSE: We thank the referee for their continued support.

Reviewer #2 (Remarks to the Author):

The authors have addressed my concerns. The manuscript should be published in Nature Communications.

I have a minor comment: the second sentence in the abstract should be rewritten, there is something wrong with the wording of the second clause. Should it be , "but this range sits between ionic lanthanide and ..."?

RESPONSE: Fair point, we have revised the sentence. We thank the referee for their support.

---End---